# Vitamin D Concentrations at Term Do Not Differ in Newborns and Their Mothers with and without Polycystic Ovary Syndrome

**DOI:** 10.3390/jcm10030537

**Published:** 2021-02-02

**Authors:** Martina Kollmann, Barbara Obermayer-Pietsch, Elisabeth Lerchbaum, Sarah Feigl, Rüdiger Hochstätter, Gudrun Pregartner, Christian Trummer, Philipp Klaritsch

**Affiliations:** 1Division of Obstetrics and Maternal Fetal Medicine, Department of Obstetrics and Gynecology, Medical University of Graz, 8036 Graz, Austria; s.feigl@medunigraz.at (S.F.); ruediger.hochstaetter@medunigraz.at (R.H.); philipp.klaritsch@medunigraz.at (P.K.); 2Division of Endocrinology and Diabetology, Department of Internal Medicine, Medical University of Graz, 8036 Graz, Austria; barbara.obermayer@medunigraz.at (B.O.-P.); elisabeth.lerchbaum@medunigraz.at (E.L.); christian.trummer@medunigraz.at (C.T.); 3Institute for Medical Informatics, Statistics and Documentation (IMI), Medical University of Graz, 8036 Graz, Austria; gudrun.pregartner@medunigraz.at

**Keywords:** polycystic ovary syndrome (PCOS), vitamin D, 25(OH)D

## Abstract

Studies suggest that non-pregnant women with polycystic ovary syndrome (PCOS) may be at elevated risk of 25 hydroxyvitamin D (25(OH)D) deficiency. Furthermore, there is evidence suggesting that 25(OH)D may also play an important role during pregnancy. Data regarding 25(OH)D deficiency during pregnancy in PCOS patients and its association with perinatal outcome is scarce. The aim of the study was to investigate whether mothers with and without PCOS have different 25(OH)D levels at term, how maternal 25(OH)D levels are reflected in their offspring, and if 25(OH)D levels are associated with an adverse perinatal outcome. Therefore, we performed a cross-sectional observational study and included 79 women with PCOS according to the ESHRE/ASRM 2003 definition and 354 women without PCOS and an ongoing pregnancy ≥ 37 + 0 weeks of gestation who gave birth in our institution between March 2013 and December 2015. Maternal serum and cord blood 25(OH)D levels were analyzed at the day of delivery. Maternal 25(OH)D levels did not differ significantly in women with PCOS and without PCOS (*p* = 0.998), nor did the 25(OH)D levels of their respective offspring (*p* = 0.692). 25(OH)D deficiency (<20 ng/mL) was found in 26.9% and 22.5% of women with and without PCOS (*p* = 0.430). There was a strong positive correlation between maternal and neonatal 25(OH)D levels in both investigated groups (r ≥ 0.79, *p* < 0.001). Linear regression estimates of cord blood 25(OH)D levels are about 77% of serum 25(OH)D concentrations of the mother. Compared to healthy controls, the risk for maternal complications was increased in PCOS women (48% vs. 65%; *p* = 0.009), while there was no significant difference in neonatal complications (22% and 22%; *p* = 1.0). However, 25(OH)D levels were similar between mothers and infants with and without perinatal complications. Although the share of women and infants with 25(OH)D deficiency was high in women with PCOS and without PCOS, it seems that the incidence of adverse perinatal outcome was not affected. The long-term consequences for mothers and infants with a 25(OH)D deficiency have to be investigated in future studies.

## 1. Introduction

Polycystic ovary syndrome (PCOS) is a heterogeneous endocrine disorder which affects several body systems and leads to reproductive and metabolic complications [1,2,3,4,5]. We previously demonstrated that the risk for maternal complications during pregnancy was increased in PCOS patients as compared with healthy controls [4]. Complications include gestational diabetes, pregnancy-induced hypertension (PIH), pre-eclampsia, preterm delivery, and an increased risk for caesarean section [4,5,6,7]. The offspring of PCOS women, on the other hand, is affected by higher intensive care unit (ICU) admission rates and lower birth weight [7]. The underlying etiology remains unclear; however, obesity, maternal hyperandrogenism, insulin resistance, and metabolic and hormonal abnormalities, which appear to vary across PCOS phenotypes, could play a crucial role [5]. 

There is some evidence showing that low serum 25 hydroxyvitamin D (25[OH]D) levels may be additional risk factors for hypertensive disorders in pregnancy such as pre-eclampsia, PIH, and gestational diabetes [8,9,10,11,12,13,14,15,16,17]. 25(OH)D deficiency is more frequent in PCOS patients, at least in non-pregnant cohorts [18,19]. It was, therefore, suspected that 25(OH)D deficiency may also be frequently present in pregnant women with PCOS and add to the increased rate of perinatal complications in this population. Since the mother is the only source of 25(OH)D for the fetus, a high correlation of maternal and cord blood 25(OH)D concentration is usually found. One study suggests that cord blood 25(OH)D levels are about 50 to 80% of serum 25(OH)D concentrations of the mother [20]. Data suggest that in pregnancy, the role of the 25(OH)D system becomes particularly important for immunomodulation of the maternal-fetal interface and that low serum 25(OH)D levels are associated with some adverse perinatal outcomes [21,22,23,24,25,26,27,28,29]. 

Herein, we aimed to investigate whether mothers with and without PCOS have different 25(OH)D levels at term and how 25(OH)D levels in PCOS patients and their offspring are associated with perinatal outcome.

## 2. Materials and Methods

### 2.1. Study Design

This cross-sectional observational study was performed at a single academic tertiary hospital (Department of Obstetrics and Gynecology, Medical University of Graz, Austria) between March 2013 and December 2015. 

### 2.2. Ethical Approval

The institutional review board approved the study (ethics committee at the Medical University of Graz, Austria; 24-179ex11/12) and written informed consent was provided by all participants.

### 2.3. Participants

Women with PCOS according to the ESHRE/ASRM 2003 definition [30] and women without PCOS and an ongoing pregnancy ≥ 37 + 0 weeks of gestation were invited to participate. Diagnosis of PCOS was previously described [31]. Only singleton pregnancies were included, and patients with severe comorbidities (neurodegenerative, immune mediated, cardiovascular, or infectious disease), suspected abnormal placentation (placenta accreta, increta, or percreta), placenta previa, previous vertical uterine incision, a history of major abdominal surgery, or known fetal malformations were excluded.

### 2.4. Outcome Measures

Primary outcome parameters were maternal and neonatal (i.e., cord blood) 25(OH)D levels at delivery. Secondary outcome parameter was the perinatal complication rate in women and newborns and its association with maternal and neonatal 25(OH)D levels. Perinatal complication was a binary composite outcome consisting of gestational diabetes, PIH, pre-eclampsia, or operative delivery for the mother and small for gestational age, large for gestational age, fetal growth restriction, fetal acidosis, or intensive care unit admission for the newborn. Data on androgens and Anti-Mullerian Hormone (AMH) concentrations in the same population were recently published [31].

### 2.5. Data Sources/Measurement

Blood samples were collected from the mother and neonate within the first 5 min after delivery. For the neonate mixed umbilical cord blood was used. Laboratory kits and assays did not change from 2013 to 2015. 25(OH)D (normal range 30–60 ng/mL) was measured daily based using a commercially available ELISA (Immunodiagnostic Systems GmbH, Frankfurt am Main, Germany) with intra- and inter-assay coefficients of variation (CV) of 5.6 and 6.4%, respectively.

Definitions used of gestational diabetes, PIH, pre-eclampsia, operative delivery, small for gestational age, large for gestational age, fetal growth restriction, and fetal acidosis have been previously described [4]. 25(OH)D deficiency was defined as levels < 20 ng/mL [32]. 

### 2.6. Sample Size

It was estimated that 350 non-PCOS and 35 PCOS patients were sufficient to detect differences between the two groups for effect sizes of at least 0.5 with a significance level of 5% and a power of 80%. A data quality check after one year of recruitment revealed that more patients than expected had to be excluded due to comorbidities and that cord blood analysis was not feasible in some cases due to insufficient material. We therefore planned to recruit at least 80 PCOS patients and 420 non-PCOS patients.

### 2.7. Statistical Methods

The current analysis is a cross-sectional analysis of a previously published prospective cohort study 31. For categorical variables, absolute and relative frequencies are reported, whereas continuous variables are expressed as median with minimum and maximum. Group differences were assessed using Fisher’s exact test for categorical variables and Mann–Whitney U test for continuous variables. To assess the association between maternal and neonatal 25(OH)D levels, a Pearson correlation analysis and linear regression analysis were performed. All analyses were performed using the statistic software R (version 3.5.3, Vienna, Austria) [33]. A *p*-value < 0.05 was considered to be statistically significant. The results of any subgroup analysis should be interpreted in an exploratory fashion. 

## 3. Results

### 3.1. Participants

In total, 499 pregnant women were assessed for eligibility and 433 were finally included for analysis (79 women with PCOS according to ESHRE/ASRM 2003 definition and 354 non-PCOS women). A flow chart of assessed and included patients as well as demographic data was previously published [31]. Prenatal vitamin intake in both groups was similar (45.6% vs. 47.5%). All taken supplements contained 25(OH)D with a maximum dose of 400 IU per day.

### 3.2. Primary Results

25(OH)D was measurable in 84.8% (67/79) of women with PCOS and in 89.3% (316/354) of Non-PCOS women. Neonatal samples were available in 27 PCOS girls (75%), 151 Non-PCOS girls (84%), 29 PCOS boys (67.4%), and 149 Non-PCOS boys (84.7%). 

Mean maternal 25(OH)D level was 29.7 ng/mL (range 9.8–57.3) in PCOS women and 29.72 ng/mL (7.3–68.2) in non-PCOS women. Levels were comparable (*p* = 0.998). Mean 25(OH)D levels in neonates born to PCOS women (25.64 ng/mL (7.0–52.9)) were comparable to mean 25(OH)D levels from neonates born to non-PCOS women (24.99 ng/mL (6.2–78.3); *p* = 0.692). Mean 25(OH)D levels from girls born to PCOS women were 26.33 ng/mL (7.0–50.0) and 24.21 ng/mL (7.0–66.6) from girls born to non-PCOS women. Those levels did not differ significantly (*p* = 0.286). Mean 25(OH)D levels from boys born to PCOS women (24.99 ng/mL (8.6–52.9)) were comparable to mean 25(OH)D levels from boys born to non-PCOS women (25.77 ng/mL (6.2–78.3); *p* = 0.594). 

The share of 25(OH)D deficiency (<20 ng/mL) was high but comparable in all investigated groups (Table 1).

There was a strong correlation between maternal and neonatal 25(OH)D levels in both groups (r ≥ 0.79, *p* < 0.001) (Figure 1). Using linear regression to estimate neonatal from the maternal 25(OH)D levels, we obtained a regression line of 0.77 × maternal level + 2.28. This means that cord blood 25(OH)D levels are about 77% of serum 25(OH)D concentrations of the mother. We did not find higher deviations from the maternal–neonatal pairs of values with higher 25(OH)D levels.

### 3.3. Secondary Results

Compared to healthy controls, the risk for maternal complications was increased in PCOS women (51 (64.6%) vs. 171 (48.3%); *p* = 0.009), while there was no significant difference in neonatal complications (17 (21.5%) vs. 79 (22.3%); *p* = 1.0). 

Women with PCOS had a higher risk of developing gestational diabetes (12 (15.2%) vs. 21 (5.9%); *p* = 0.009) when compared to healthy controls. No significant difference was found for PIH (8 (10.1%) vs. 17 (4.8%); *p* = 0.103), pre-eclampsia (3 (3.8%) vs. 5 (1.4%); *p* = 0.164), and operative delivery (41 (51.9%) vs. 148 (41.8%); *p* = 0.259).

Large (3 (3.8%) vs. 19 (5.4%); *p* = 0.779) or small (9 (11.4%) vs. 44 (12.4%); *p* = 1.0) for gestational age (SGA) neonates, fetal growth restriction (1 (1.3%) vs. 3 (0.8%); *p* = 0.555), fetal acidosis (2 (2.5%) vs. 10 (2.8%); *p* = 1.0), and ICU admission rate (1 (1.3%) vs. 5 (1.4%); *p* = 1.0) occurred with a comparable frequency in both groups.

The prevalence of adverse maternal and neonatal outcomes was not associated with 25(OH)D levels, with one exception: 25(OH)D levels were higher in PCOS offspring with fetal acidosis compared to 25(OH)D levels from PCOS offspring without fetal acidosis (Table 2).

25(OH)D deficiency was high in all groups (Table 1). However, we did not find an association between 25(OH)D deficiency and adverse perinatal outcome (Table 3). 

## 4. Discussion

### 4.1. Key Results

Maternal 25(OH)D levels did not differ significantly in women with PCOS and without PCOS, nor did the 25(OH)D levels of their offspring. The share of 25(OH)D deficiency was high in all investigated groups. Cord blood 25(OH)D levels were about 77% of serum 25(OH)D concentrations of the mother. Compared to healthy controls, the risk for maternal complications was increased in PCOS women, while there was no significant difference in neonatal complications. However, 25(OH)D levels were similar between mothers and infants with and without perinatal complications.

### 4.2. Interpretation

While we found comparable 25(OH)D levels between mothers and infants with and without perinatal complications and between women with PCOS and without PCOS, an adequate vitamin supply to pregnant women is lacking in nearly 40% of pregnancies. Our findings confirm that 25(OH)D deficiency is common in pregnant women with PCOS and without PCOS and even more frequently found in their offspring as cord blood 25(OH)D levels are only 77% of serum 25(OH)D concentrations of the mothers. A sufficient 25(OH)D supplementation can prevent neonatal hypocalcemia, which may result in softening of bones and can lead to seizures and dilated cardiomyopathy 21, [34]. It has further been shown, that a preconceptional 25(OH)D supplementation is advantageous for women with PCOS and can further have an positive impact on folliculogenesis [35,36,37]. The current guidelines recommend a 25(OH)D intake of 400 to 800 IU per day for pregnant women; however, the circulating 25(OH)D concentration that is sufficient to meet the physiological needs of humans is an ongoing subject of debate 21, [38,39,40]. A randomized controlled trial (RCT) published by O’Callaghan et al. has shown that an overall 25(OH)D intake of almost 1200 IU of 25(OH)D per day was necessary to ensure that cord blood serum 25(OH)D concentrations were above 30 nmol/L in 95%, and above 25 nmol/L in 99% of infants, respectively [20]. The recommended upper limit of intake is 4000 IU per day 39, 40. Pilz et al. in a recently published review recommend an intake of vitamin D of 800 to 1000 IU per day during preconception or pregnancy to achieve serum 25(OH)D target concentrations as recommended by vitamin D guidelines 21. Prenatal vitamin supplements usually contain only 200 to 400 IU and, therefore, seem to be an insufficient source of 25(OH)D.

Our study has limitations that should be noted. We used an assay which measured total vitamin D (D2 and D3) instead of only vitamin D3, the main active compound. This fact might influence our results.

Regarding perinatal complications there are several RCTs and meta-analyses of RCTs which evaluated the effect of vitamin D supplementation during pregnancy on neonatal and maternal outcome [41,42,43,44,45,46,47,48,49,50,51,52,53,54,55]. A recently published meta-analysis included 43 trials with 8406 participants and evaluated effects of vitamin D supplementation on 11 maternal and 27 neonatal outcomes 54. However, due to missing data on clinical outcome in many cases, most analyses were based on only a minority of trials. This meta-analysis found that vitamin D supplementation increases mean birth weight by 58.33 g, reduces the risk of SGA and the risk of asthma or wheeze in the offspring by 3 years of age. It did not find effects on preterm birth, pre-eclampsia, neonatal death, caesarean section, preterm labor or infections 54. A further meta-analysis investigated the effect of vitamin D supplementation on neonatal outcomes and found similar results 55. Vitamin D supplementation decreased the risk of SGA and increased birth weight. However, there was no significant effect on asthma 55. Interestingly, Apgar scores at 1-min increase by 0.09 and at 5 min by 0.08. None of the published studies reported an association of 25(OH)D levels and fetal acidosis 55. Summarizing the available evidence on vitamin D supplementation in pregnancy it can be concluded that vitamin D supplementations is safe and improves vitamin D and calcium status. Data on neonatal and maternal outcome are for the bigger part still inconsistent 54, 55. The benefits of vitamin D supplementation before and during pregnancy should be further investigated through thoroughly planned intervention studies, and it is crucial to define ethnicity, season and period of supplementation.

## Figures and Tables

**Figure 1 jcm-10-00537-f001:**
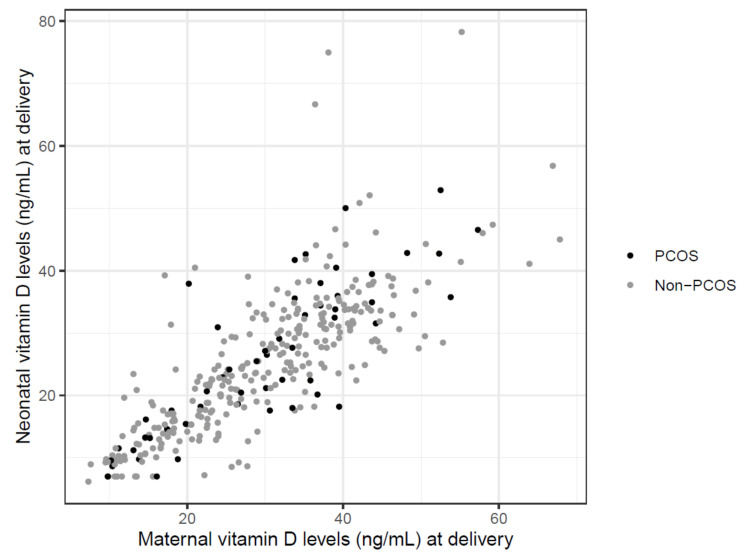
To assess the association between maternal and neonatal 25(OH)D levels, a Pearson correlation analysis and linear regression analysis were performed and a strong correlation between maternal and neonatal 25(OH)D levels in both groups (r ≥ 0.79, *p* < 0.001) was found.

**Table 1 jcm-10-00537-t001:** 25(OH)D deficiency (<20 ng/mL); variables were analyzed by using Fisher’s exact test.

	25(OH)D Deficiency		*p*-Value
	**n**	**%**	
PCOS women	18	26.9	0.43
Non-PCOS women	71	22.5	
Neonates of PCOS women	21	37.5	0.88
Neonates on Non-PCOS women	107	35.7	
PCOS girls	10	37	1.0
Non-PCOS girls	58	38.4	
PCOS boys	11	37.9	0.70
Non-PCOS boys	49	32.9	

PCOS: Polycystic Ovary Syndrome.

**Table 2 jcm-10-00537-t002:** Association between 25(OH)D levels and maternal and neonatal complications in PCOS and Non- PCOS pregnancies.

	PCOS Women				*p*-Value	Non-PCOS Women				*p*-Value
Maternal Complications	Yes		No			Yes		No		
Gestational diabetes	30.4	14.6–53.8	30.2	9.8–57.3	0.735	34.5	11.9–66.9	28.8	7.3–68.2	0.190
PIH	30.4	16.2–48.2	30.2	9.8–57.3	0.790	23.6	11.9–40.1	29.8	7.3–68.2	0.128
Pre-eclampsia	32.8	30.4–35.2	30.1	9.8–57.3	0.658	30.4	11.9–34.1	29.1	7.3–68.2	0.476
Operative delivery	30.5	10.4–57.3	30.1	9.8–52.5	0.320	29.4	7.6–67.8	29.3	7.3–68.2	0.871
Any complication	32.0	10.4–57.3	28.9	9.8–52.5	0.132	29.1	7.6–67.8	29.8	7.3–68.2	0.622
	**Neonates of PCOS Women**				***p*-Value**	**Neonates of Non-PCOS Women**				***p*-Value**
**Neonatal Complications**	**Yes**		**No**			**Yes**		**No**		
SGA (<10th percentile)	21.3	15.4–39.4	24.8	7.0–52.9	0.832	29.6	10.3–56.8	23.8	6.2–78.3	0.082
LGA (>90th percentile)	24.8	14.6–34.9	23.5	7.0–52.9	0.912	19.8	7.0–37.7	24.6	6.2–78.3	0.213
Fetal growth restriction	-	-	23.5	7.0–52.9	-	46.0	35.1–56.8	24.3	6.2–78.3	0.039
Fetal acidosis	44.1	41.7–46.5	22.7	7.0–52.9	0.040	27.1	13.9–46.1	24.3	6.2–78.3	0.394
ICU	34.9	34.9–34.9	22.9	7.0–52.9	0.421	37.5	35.1–46.1	24.2	6.2–78.3	0.015
Any complication	32.2	14.6–46.5	22.7	7.0–52.9	0.219	27.6	7.0–56.8	23.8	6.2–78.3	0.199

Data are presented as median (range) and differences were assessed by Mann–Whitney U test. PIH = Pregnancy induced hypertension, Operative delivery = elective and non-elective caesarean section and operative vaginal delivery; SGA = small for gestational age; ICU = intensive care unit admission. Total maternal complication rate = gestational diabetes, pregnancy-induced hypertension, pre-eclampsia, non-elective caesarean section, and operative vaginal delivery, preterm birth. Total neonatal complication rate = large or small for gestational age, acidosis, ICU admission, and pre- and perinatal mortality.

**Table 3 jcm-10-00537-t003:** Association between 25(OH)D deficiency and maternal and neonatal complications in PCOS and Non-PCOS pregnancies.

Maternal complications	PCOS women 25(OH)D < 20 ng/mL	PCOS women 25(OH)D ≥ 20 ng/mL	*p*-value
	11 (61.1)	33 (67.3)	0.773
	Non-PCOS women 25(OH)D < 20 ng/mL	Non-PCOS women 25(OH)D ≥ 20 ng/mL	
	34 (47.9)	121 (49.4)	0.893
Neonatal complications	Neonates of PCOS women 25(OH)D < 20 ng/mL	Neonates of PCOS women 25(OH)D ≥ 20 ng/mL	
	3 (14.3)	9 (25.7)	0.503
	Neonates of Non-PCOS women 25(OH)D < 20 ng/mL	Neonates of Non-PCOS women 25(OH)D ≥ 20 ng/mL	
	19 (17.8)	43 (22.3)	0.376

Data are presented as n (%) and differences were assessed by Fisher’s exact test. Maternal complications = gestational diabetes, pregnancy-induced hypertension, pre-eclampsia, non-elective caesarean section, and operative vaginal delivery, preterm birth. Neonatal complications = large or small for gestational age, acidosis, ICU admission, and pre- and perinatal mortality.

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
