# Peer review of "Vitamin D Concentrations at Term Do Not Differ in Newborns and Their Mothers with and without Polycystic Ovary Syndrome"

_jcm, 2021, doi:10.3390/jcm10030537_

Round 1

Reviewer 1 Report

The main aims of the study were to determine if:

  1. Vitamin D levels were different in subjects with and without PCOS – 79 PCOS subjects are far too few to determine this outcome in a population that largely vitamin D sufficient
  2. How 25(OH)D levels in PCOS patients and their offspring are associated with perinatal outcome – again there were too few PCOS patients with vitamin D deficiency to realistically exclude a type 2 statistical error.

Comments in addition to those noted above

  1. The study is said to be prospective but it would seem to be cross sectional observational if women were recruited essentially at the time of delivery (>37+0 weeks)
  2. Given the reported vitamin D deficiency prevalence in PCOS this population appeared different – was the sample biased in some way?
  3. Sample size – on what was the sample size estimated from – as it is written it does not make sense. Furthermore, if 35 PCOS patients were sufficient to detect a difference (in what?)  and there were only 18 with vitamin D deficiency then this would suggest the study was underpowered?
  4. The details of the vitamin D assay need to including CVs
  5. How many were taking vitamin supplements including vitamin D?
  6. The assay measured total vitamin D (D2 and D3) but it has been suggested that vitamin D3 is the main active compound. This needs to be discussed

Author Response

Thank you very much for considering our manuscript entitled “Vitamin D concentrations at term do not differ in newborns and their mothers with and without polycystic ovary syndrome” suitable for publication in Journal of Clinical Medicine. We took care to respond to the points raised by the reviewers and adopted the manuscript according to the suggestions.

Response to Reviewer 1 Comments:

The main aims of the study were to determine if:

  • Vitamin D levels were different in subjects with and without PCOS – 79 PCOS subjects are far too few to determine this outcome in a population that largely vitamin D sufficient

We agree with the issue of limited sample size. However, no study on this issue is available so far. We therefore think that our study still offers important clinical information and may serve as the basis for larger multicenter studies.

  • How 25(OH)D levels in PCOS patients and their offspring are associated with perinatal outcome – again there were too few PCOS patients with vitamin D deficiency to realistically exclude a type 2 statistical error.

Thank you for this valuable comment. As mentioned above, we agree with the issue of limited sample size. However, no study on this issue in PCOS patients is available so far. We therefore think that our study still offers important clinical information and may serve as the basis for larger multicenter studies.

Comments in addition to those noted above

  • The study is said to be prospective but it would seem to be cross sectional observational if women were recruited essentially at the time of delivery (>37+0 weeks)

Thank you for this valuable comment. We adopted the section as suggested.

  • Given the reported vitamin D deficiency prevalence in PCOS this population appeared different – was the sample biased in some way?

Thank you for this valuable comment. We are aware that the selected study type cannot fully prevent bias, however we took care to avoid any bias, including selection bias. Demographical data was similar in both groups.

  • Sample size – on what was the sample size estimated from – as it is written it does not make sense. Furthermore, if 35 PCOS patients were sufficient to detect a difference (in what?)  and there were only 18 with vitamin D deficiency then this would suggest the study was underpowered?

We report on a cross-sectional analysis of a previously published prospective cohort study. The initial sample size estimation states that for any comparison between PCOS and non-PCOS patients the power to detect an effect size (for example Cohen's d, defined as the difference between two means divided by a standard deviation) of at least 0.5 is 80%, irrespective of the parameter investigated. However, the results of any subgroup analyses should be interpreted in an exploratory fashion. To make this clear we added respective sentences in the statistical methods section.

  • The details of the vitamin D assay need to including CVs

Thank you for this comment. We added the following paragraph: “[…] with intra- and inter-assay coefficients of variation (CV) of 5.6 and 6.4% respectively.”

  • How many were taking vitamin supplements including vitamin D?

Thank you for this valuable comment. We added the following information: “Prenatal vitamin intake in both groups was similar (45.6% versus 47.5%). All taken supplements contained 25(OH)D with a maximum dose of 400 IU per day.”

  • The assay measured total vitamin D (D2 and D3) but it has been suggested that vitamin D3 is the main active compound. This needs to be discussed

Thank you for this valuable comment. To address this issue, we included the following paragraph in the discussion: "Our study has limitations that should be noted. We used an assay which measured total vitamin D (D2 and D3) instead of only vitamin D3, the main active compound. This fact might influence our results.”

Reviewer 2 Report

In the manuscript entitled „Vitamin D concentrations at term do not differ in newborns 3 and their mothers with and without polycystic ovary syndrome” the Authors investigate whether the level of vitamin D can be associated with perinatal outcomes in PCOS and non-PCOS women. The study is technically sound. The manuscript is generally well-written. In my opinion only a minor, optional revision can be made.

The obtained results are quite surprising since no significant differences between the maternal levels of 25(OH)D in women with PCOS and without PCOS. The offspring’s 25(OH)D levels was also similar in both examined groups.  Thus, the risk of neonatal Vitamin D deficiency is similar in both groups.

As could be anticipated, the risk for maternal complications was increased in PCOS women when compared to healthy controls. The authors also concluded that there was no significant difference in neonatal complications.

In my opinion, the Authors may discuss the fact that the vitamin D supplementation is advantageous for the women with PCOS before the pregnancy. Since according to the literature a greater thickness of the endometrium was found in women with PCOS who had normal levels of vitamin D, which resulted in a greater chance of getting pregnant [Lerchbaum, E.; Rabe, T. Vitamin D and female fertility. Curr. Opin. Obstet. Gynecol. 2014, 26, 145–150.]. Additionally, vitamin D attenuates the effects of advanced glycation end products in women with PCOS (enhanced androgen synthesis, abnormal folliculogenesis) [Merhi, Z.; Buyuk, E.; Cipolla, M. Advanced glycation end products alter steroidogenic gene expression by granulosa cells: An effect partially reversible by vitamin D. MHR Basic Sci. Reprod. Med. 2018, 24, 318–326., Merhi, Z. Crosstalk between advanced glycation end products and vitamin D: A compelling paradigm for the treatment of ovarian dysfunction in PCOS. Mol. Cell. Endocrinol. 2019, 479, 20–26.]. From this perspective, one may ponder whether the results described in the submitted manuscript refer to the group of PCOS patients who mostly had a normal vitamin D level  before getting pregnant.

Author Response

Thank you very much for considering our manuscript entitled “Vitamin D concentrations at term do not differ in newborns and their mothers with and without polycystic ovary syndrome” suitable for publication in Journal of Clinical Medicine. We took care to respond to the points raised by the reviewers and adopted the manuscript according to the suggestions.

Response to Reviewer 2 Comments:

In the manuscript entitled „Vitamin D concentrations at term do not differ in newborns 3 and their mothers with and without polycystic ovary syndrome” the Authors investigate whether the level of vitamin D can be associated with perinatal outcomes in PCOS and non-PCOS women. The study is technically sound. The manuscript is generally well-written. In my opinion only a minor, optional revision can be made.

The obtained results are quite surprising since no significant differences between the maternal levels of 25(OH)D in women with PCOS and without PCOS. The offspring’s 25(OH)D levels was also similar in both examined groups.  Thus, the risk of neonatal Vitamin D deficiency is similar in both groups.

As could be anticipated, the risk for maternal complications was increased in PCOS women when compared to healthy controls. The authors also concluded that there was no significant difference in neonatal complications.

In my opinion, the Authors may discuss the fact that the vitamin D supplementation is advantageous for the women with PCOS before the pregnancy. Since according to the literature a greater thickness of the endometrium was found in women with PCOS who had normal levels of vitamin D, which resulted in a greater chance of getting pregnant [Lerchbaum, E.; Rabe, T. Vitamin D and female fertility. Curr. Opin. Obstet. Gynecol. 2014, 26, 145–150.]. Additionally, vitamin D attenuates the effects of advanced glycation end products in women with PCOS (enhanced androgen synthesis, abnormal folliculogenesis) [Merhi, Z.; Buyuk, E.; Cipolla, M. Advanced glycation end products alter steroidogenic gene expression by granulosa cells: An effect partially reversible by vitamin D. MHR Basic Sci. Reprod. Med. 2018, 24, 318–326., Merhi, Z. Crosstalk between advanced glycation end products and vitamin D: A compelling paradigm for the treatment of ovarian dysfunction in PCOS. Mol. Cell. Endocrinol. 2019, 479, 20–26.]. From this perspective, one may ponder whether the results described in the submitted manuscript refer to the group of PCOS patients who mostly had a normal vitamin D level  before getting pregnant.

Thank you for this valuable comment. We added the suggested points to the discussion section.